# A Screening for Virus Infections among Wild Eurasian Tundra Reindeer (*Rangifer tarandus tarandus*) in Iceland, 2017–2019

**DOI:** 10.3390/v15020317

**Published:** 2023-01-23

**Authors:** Morten Tryland, Javier Sánchez Romano, Ingebjørg Helena Nymo, Torill Mørk, Rán Þórarinsdóttir, Eva Marie Breines, Hong Li, Cristina Wetzel Cunha, Skarphéðinn G. Thórisson

**Affiliations:** 1Department of Arctic and Marine Biology, UiT The Arctic University of Norway, Framstredet 39, Breivika, N-9019 Tromsø, Norway; 2Department of Forestry and Wildlife Management, Inland Norway University of Applied Sciences, N-2480 Koppang, Norway; 3Vascular Biology Research Group, Department of Medical Biology, UiT The Arctic University of Norway, N-9019 Tromsø, Norway; 4Norwegian Veterinary Institute, Holtveien 66, N-9016 Tromsø, Norway; 5East Iceland Nature Research Centre, Tjarnarbraut 39B, 700 Egilsstaðir, Iceland; 6Animal Disease Research Unit, US Department of Agriculture—Agricultural Research Service, Washington State University, Pullman, WA 99164, USA

**Keywords:** alphaherpesvirus, bluetongue virus, gammaherpesvirus, MCFV, parapoxvirus, pestivirus, schmallenbergvirus, serology, viral pathogens, wildlife

## Abstract

A winter population of around 4000–5000 wild Eurasian tundra reindeer (*Rangifer t. tarandus*) in the eastern part of Iceland represents descendants from 35 semi-domesticated reindeer imported to Iceland from Finnmark county, Norway, in 1787. While previous studies have indicated that they host fewer parasite species as compared to reindeer in Fennoscandia, little information exists on their exposure to reindeer viral pathogens. The aim of this study was to investigate blood from hunted reindeer for antibodies against alphaherpesvirus and gammaherpesviruses (malignant catarrhal fever viruses, MCFV), pestivirus, bluetongue virus, and Schmallenberg virus, and to investigate nasal and oral mucosal membrane swab samples for the presence of parapoxvirus-specific DNA. Blood samples collected during the hunting seasons in 2017 (*n* = 40), 2018 (*n* = 103), and 2019 (*n* = 138) were tested for viral antibodies using enzyme-linked immunosorbent assays (ELISA). Screening for parapoxvirus DNA was conducted on swab samples from 181 reindeer by polymerase chain reaction (PCR), targeting the *B2L* and *GIF* genes. Antibodies against pestivirus were detected in two animals from 2017, and antibodies against MCFV were detected in two reindeer from 2018. No antibodies were detected against the other viruses tested. Parapoxvirus-specific DNA was detected in nasal swab samples from two animals sampled in 2019. This study suggests that the investigated viral infections are either not present or present at a low prevalence only, probably not representing a major health threat to this reindeer population. The lack of exposure to alphaherpesvirus, an enzootic pathogen in most investigated *Rangifer* populations, was unexpected.

## 1. Introduction

Reindeer and caribou (*Rangifer tarandus*), often called *Rangifer*, consist of several subspecies that are distributed throughout the Arctic region, and the importance of these animals, representing food, fur and other valuable materials, is reflected among many peoples and cultures. The majority of the *Rangifer* in North America, counting about 2,000,000–3,000,000 animals, are wild, and several populations have experienced dramatic reductions in the past few decades [1,2]. Most of the 3,000,000–4,000,000 wild and semi-domesticated reindeer in Eurasia belong to the Eurasian tundra reindeer subspecies, *Rangifer tarandus tarandus* [2]. In Fennoscandia today, reindeer herding is based on semi-domesticated tundra reindeer, and represents an important livelihood and cultural corner stone of the Sami people [3].

Eurasian tundra reindeer (*Rangifer tarandus tarandus*) were introduced from Norway to Iceland on four occasions; in 1771 (*n* = 13–14, extinct around 1783), 1777 (*n* = 30, extinct around 1920–1930), 1784 (*n* = 30–35, probably extinct around 1937), and finally, in 1787, when 30 female reindeer and five reindeer bulls were gathered in Avjovarre, Finnmark, Norway, and shipped to Vopnafjördur on the north-east coast of Iceland [4]. This latter introduction was successful. Aerial surveys to assess population size, sex, and age composition of the population are conducted each year in July and during the rut in the fall [5]. In 2000, the total population (summer stock, i.e., after calving, before harvest) was estimated to consist of about 3000 animals. The population doubled from the year 2000 to 2008 but has since then been stable or has slightly decreased [6].

Although the original purpose of introducing reindeer to Iceland was to establish reindeer herding [4], the animals have since the introduction been free-ranging and are still managed as a wild population. The reindeer population is today distributed in the north-east and south-east of Iceland [5], and the population is divided into 9 different management zones (Area 1–9).

Although there are a few reports that the arctic fox (*Vulpes lagopus*), which is the only non-introduced terrestrial mammal in Iceland, may prey on newborn reindeer calves [7] there are no reindeer predators in Iceland. Thus, the main tool for regulating the population is hunt, which is conducted through the purchase of a license for a bull or a female, and with the requisite of the hunt being associated with a licensed hunting guide. Calves are normally not hunted. The hunting quotas for each management zone are based on the annual counts and population estimates. For the three hunting seasons during the project period (2017–2019), the total reindeer population, after calving and before hunt, consisted of 6000–7000 animals, of which a total of 4177 reindeer were hunted, approximately 1300–1400 animals per year. Of the hunted reindeer, 71% were females [6,8,9].

The hunt is considered the most important mortality factor for Icelandic reindeer. Other mortality factors, as indicated through the investigations of 252 reindeer carcasses (1991–2013; Snæfellsherd, Area 1 and 2), were car accidents (≈40%), and other types of traumas associated with rut, drowning, illegal hunt, calving complications, and snow avalanche (≈ 35%), leaving approximately 25% with unknown cause of death [10].

Several viruses that are pathogenic to reindeer and may be relevant for the wild Icelandic reindeer have been identified in the Fennoscandian population of semi-domesticated reindeer, including reindeer in Finnmark County, Norway, from where the Icelandic reindeer originated in 1787.

The reindeer alphaherpesvirus (family *Herpesviridae*, genus *Varicellovirus*), Cervid herpesvirus 2 (CvHV2), or a similar virus, is enzootic in most studied reindeer and caribou populations [11,12], including the Fennoscandian semi-domesticated reindeer [13,14,15]. CvHV2 is known to cause infectious keratoconjunctivitis (IKC) [16,17] and to facilitate secondary bacterial infections in semi-domesticated reindeer in Fennoscandia [18].

Gammaherpesviruses (family *Herpesviridae*; genus *Macavirus*) within the malignant catarrhal fever virus (MCFV) group causes MCF in domestic and wild ruminants and can also infect reindeer [19]. A serological study of semi-domesticated reindeer in Finnmark County, Norway, revealed MCFV antibodies in 3.5% of 3339 apparently healthy reindeer sent for slaughter [20]. Furthermore, MCF has been diagnosed in an adult male from a herd of semi-domesticated reindeer in Norway, displaying characteristic clinical signs such as corneal opacity and fibrinopurulent eye flood, thickening of the skin with crusts, and the finding of mononuclear cells around blood vessels and necrotizing vasculitis in the CNS [19].

Similarly, and among the same animals from Finnmark, Norway, that were screened for MCFV antibodies, a prevalence of antibodies against Pestivirus (*Flaviviridae* family, genus *Pestivirus*) of 13% was found [21]. In a recent report investigating eight different herds of semi-domesticated reindeer in Norway, antibodies against CvHV2 and MCFV were found in all herds, with a mean prevalence of 42% (*n* = 570) and 11% (*n* = 550), respectively. A total of 570 reindeer were screened for antibodies against pestivirus, revealing seropositivity in five of the herds with a total prevalence of 19% among 400 animals [15].

These investigations have revealed exposure to MCFV and Pestivirus, but as with other free ranging animals, it may be challenging to pinpoint the exact impact such infections may have on the individual and on population level. While clinical disease and disease outbreaks are often registered, more subtle and long-term effects on reproduction, body condition, fitness, and calf survival may be harder to address.

Parapoxvirus (i.e., Orf virus, ORFV; Pseudocowpoxvirus, PCPV) has caused the disease contagious ecthyma (CE) in reindeer in Finland, Sweden, and Norway [19], and the disease also appeared in at least 31 reindeer in Iceland in 2016 [22]. Parapoxvirus may either occur in one or a few animals or cause regular outbreaks, with hundreds of animals affected, as seen in Finland [23,24]. Parapoxvirus is believed to be enzootic in the sheep population in Iceland and has also caused zoonotic infections [25].

In contrast to the viruses mentioned above, there are no records of wild or semi-domesticated reindeer being exposed to bluetongue virus (BTV; family *Reoviridae*, genus *Orbivirus*) or Schmallenberg virus (SBV; family *Bunyaviridae*, genus *Orthobunyavirus*). BTV, which is being transmitted by biting midges (*Culicoides* spp.), appeared in Scandinavia in 2007–2009. It can cause a non-contagious disease in immunologically naïve sheep, but also other animals, including wild ruminants, are susceptible and regarded as important in the epidemiology of BTV [26]. SBV was first identified in Germany in 2011, causing congenital malformations in cattle and sheep and fever, diarrhea, and decreased milk production [27]. Biting midges (*Culicoides* spp.) and mosquitos transmit SBV. A serological screening of Eurasian tundra reindeer in German zoos indicated exposure of reindeers to both BTV (3.4%) and SBV (59%), indicating the susceptibility of reindeer to these infections [28].

The reindeer population in Iceland is generally regarded as healthy. In spite of several investigations revealing the presence of parasites that are known to be hosted by reindeer in other countries, none of the investigations so far conducted in Iceland conclude that these parasites are representing specific health threats to the population [29,30,31,32,33]. Except for the recently recorded outbreak of CE, there are no records on outbreaks of viral diseases among Icelandic reindeer. As part of a broader study, including reindeer from Iceland (*n* = 49), Norway, Sweden, Finland, and Russia, Reindeer papillomavirus was detected by next generation sequencing (NGS) in a nasal swab pool [34], but no thorough virological investigations for reindeer pathogens in this wild reindeer population have been reported.

The aim of this study was to conduct a serological survey for antibodies against potential viral pathogens among Icelandic reindeer. At the same time, we wanted to investigate if parapoxvirus (e.g., ORFV) was circulating among Icelandic reindeer.

## 2. Material and Methods

### 2.1. Animals and Sampling

Reindeer were sampled during the hunting seasons (15 July–20 September) in the period 2017–2019, with all the nine management zones (hunting areas) represented (Table 1).

About twice as many reindeer were sampled from Area 2 (*n* = 106) as compared to the much larger Area 1 (*n* = 58) (Figure 1, Table 1). This is due to a higher animal density in Area 2 and the fact that the population in Area 2 increased from approximately 1300 to about 1900 during the course of this study, hence increased hunting quotas and availability of samples during the hunting season [6,8,9].

Blood samples were collected by the hunter or guide in the location where the animals were shot and bled. Blood was collected directly into vacutainer tubes (BD Vacutainer^®^; BD, Plymouth, UK, for serum and plasma; EDTA). The blood samples, carcass, and head were brought to farm-based meat preparation facilities where additional sampling was conducted, usually during the same day or the morning after. Blood samples were centrifuged (10 min, 3000× *g*) and serum and plasma were stored at –20 °C until further analysis. From the tubes prepared for plasma collection, leucocytes (“buffy coat”) were obtained and stored at −20 °C (1–4 weeks) and then at −80 °C until analysis.

Swab samples (Applimed SA, Châtel-St-Denis, Switzerland) were obtained from the mucosa of the nose (approximately 5 cm inside the nostril) and mouth (between gingiva and the chin). The swabs were placed in 1.8 mL cryotubes with 800 μL of Eagle’s minimum essential medium (EMEM) containing antibiotics (10,000 U/mL penicillin and 10 mg/mL streptomycin; 1 mL/L of gentamicin, 50 mg/mL and 10 mL/L of amphotericin B 250 μg/mL; EMEMab 10 mL/L) and stored at −20 °C (1–4 weeks) and then at −80 °C until analysis.

The sex and age distribution for each hunting season is depicted in Figure 1.

### 2.2. Serological Tests

Serum samples were tested for antibodies against a panel of viruses (Table 2). Serum samples were investigated for alphaherpesvirus antibodies using a commercial bovine blocking enzyme-linked immunosorbent assay (ELISA). The kit has previously been validated against a virus neutralization test (VNT) for analyzing reindeer serum samples for antibodies against alphaherpesvirus [35]. Serum samples were tested in 1:2 dilution and evaluated against the bovine positive (PC) and negative (NC) control sera provided with the kit. A competition percentage (C%) was calculated by dividing the optical density (OD) of the sample (OD S) by the mean OD of the bovine negative control (OD NC): C% = OD S/OD NC. Samples were categorized as positive if C% ≤ 0.5, doubtful if 0.5 < C% ≤ 0.55, and negative if C% > 0.55.

Serum samples were investigated for the presence of specific antibodies to the MCFV group by a direct competitive-inhibition ELISA (cELISA) [36,37], applied for reindeer serum samples as described previously [20] (Table 2). The cELISA is based on the ability of a serum sample to compete with a monoclonal antibody (15-A) which is directed against a virus epitope, a 45 kDa glycoprotein of Alcelaphine herpesvirus 1 (AlHV1) that infects the blue wildebeest (*Connochaetes taurinus*), and other wildlife species [42]. This virus epitope is highly conserved among MCFVs, including viruses infecting sheep, goats, and cervids [38]. Sera were scored by optical density (OD), based on the ability of each sample to inhibit binding of the 15-A monoclonal antibody to the AlHV1 antigen. Samples (50 μL) were tested in duplicates, using mean OD to evaluate the result. The performance of a serum test sample was calculated as inhibition percentage (I%): I% = (100 − OD S × 100)/mean OD NC. A serum sample was considered positive if I% ≥ 25 and negative if I% < 25.

Serum samples were tested for pestivirus antibodies using a commercial blocking ELISA (Table 2) designed for domestic ruminants, using the p80/125 non-structural protein as antigen, which is presumably shared between all strains of Pestivirus A and B (i.e., former bovine viral diarrhea virus; BVDV-1 and BVDV-2) and D (i.e., former border disease virus; BDV) [39]. This kit and a similar ELISA based on the same antigen has previously been evaluated against VNTs when analyzing serum samples from Eurasian tundra reindeer for pestivirus antibodies [14,21,43]. The serum samples (S) were tested in a 1:10 dilution. Positive (PC) and negative (NC) controls were provided with the kit. A competition percentage (%P) was calculated for each sample (OD NC − OD S)/(OD NC − OD PC) × 100. A %P ≤ 30% was interpreted as being seronegative, 30 < %P < 50 as doubtful and ≥50% as being seropositive, in line with the manufacturer’s instructions for testing samples from bovines.

Serum samples were investigated for antibodies against BTV using a commercial competitive ELISA based on recombinant VP7 as antigen (Table 2). This antigen is conserved among different serotypes of BTV and thus suitable for testing of multiple species (e.g., cattle, sheep, goat, buffalo, and roe deer) [40,44] and has previously been used to analyze reindeer sera [15,28]. Samples (S; 50 μL of each) were tested undiluted in duplicates and evaluated against positive (PC) and negative control (NC) sera provided with the kit. Optical density was read at 450 nm. The results are expressed as competition percentage: S/N% = (OD S/OD NC) × 100. The results were classified as positive or negative in accordance with the evaluation criteria and cut-offs provided by the manufacturer, i.e., S/N% < 40 was considered positive and S/N% ≥ 40 was considered negative.

Serum samples were investigated for antibodies against SBV using a commercial competitive ELISA based on recombinant SBV nucleoprotein antigens for multispecies testing (Table 2). Similar kits have previously been used to detect SBV antibodies in moose (*Alces alces*), red deer (*Cervus elaphus*), fallow deer (*Dama dama*), roe deer (*Capreolus capreolus*) [45], and Eurasian tundra reindeer [15,28]. The serum samples (10 μL) were tested undiluted in duplicates. The OD score was read at 450 nm and the mean OD score of the sample was evaluated against the evaluation criteria and the positive (PC) and negative (NC) control sera provided by the manufacturer. For the 2017 samples, the competitive ID Screen^®^ kit for antibodies against SBV was evaluated by calculating a competition percentage (S/N%): S/N% = (OD S/OD NC) × 100. Samples were classified as positive if S/N% ≤ 40, doubtful if 40% < S/N% ≤ 50%, and negative if S/N% > 50%. For the 2018 and 2019 samples, an indirect ELISA was used, and the performance of a sample (S) was evaluated against PC and expressed as S/PC percentage: S/PC% = (OD S − OD NC)/(OD PC − OD NC) × 100. The results were considered positive if S/PC% >60, doubtful if 50% < S/PC% ≤ 60%, and negative if S/PC% ≤ 50%.

### 2.3. Polymerase Chain Reaction (PCR)

Swab samples from 95 adult reindeer (nasal) and 18 calves (nasal and oral) from 2018 and swab samples from 68 adult reindeer (nasal and oral) from 2019 were analyzed for the presence of parapoxvirus DNA. DNA was extracted from swab samples (Maxwell 16^®^ Buccal Swab LEV DNA purification kit; Promega, Madison, WI, USA). The PCR reactions were run in a Perkin Elmer GeneAmp^@^ PCR System 9700 (Perkin Elmer Corp., Shelton, CT, USA), targeting the putative viral envelope antigen (*B2L*; primers PPP1 → 5′-gtc gtc cac gag cag ct-3′ and PPP4 → 5′-tac gtg gga agc gcc tcg ct-3′) and a gene encoding a protein inhibiting the granulocyte macrophage colony stimulating factor and interleukin-2 (*GIF*; primers GIF 5 → 5′-gct cta gga aag atg gcg tg-3′, GIF 6 → 5′-gta ctc ctg gct gaa gag cg -3′), as previously described [46]. DNA isolated from a skin lesion of a domestic goat (*Capra hircus*) with CE was used as a positive control (Norwegian Veterinary Institute, Tromsø, Norway). Unused dNTP and primers were removed enzymatically from the PCR amplicons (ExoSAP-IT™; Amersham Pharmacia Biotech, Lund, Sweden) prior to sequencing (BigDye^®^ Terminator v3.1 cycle sequencing kit; Applied Biosystems, Tønsberg, Norway) in an Applied Biosystems 3130 XL Genetic Analyzer (Applied Biosystems).

DNA was extracted (Maxwell^®^ 16 LEV Buccal swab DNA kit; Promega, Madison, WI, USA) from leucocytes (“buffy coat”) from two reindeer with antibodies against MCFV and from eight seronegative reindeer. The samples were subjected to a consensus herpesvirus PCR [47] and an OvHV2 nested PCR [48] in attempts to amplify herpesvirus-specific or MCFV-specific sequences, respectively.

## 3. Results

The results are summarized in Table 3 and Table 4 and visualized in Figure 2. Detailed information regarding each animal sampled and results are presented in Table 1.

All animals investigated were sero-negative for alphaherpesvirus, BTV, and SBV. The alphaherpesvirus ELISA results (OD sample/OD negative) ranged from 0.64 to 1.23 with a mean score of 0.91 (SD: 0.083, *n* = 281). The BTV ELISA results ((OD sample/OD negative control) × 100) ranged from 72.34 to 453.98 with a mean of 131.62 (SD: 26.268, *n* = 281). The SBV ELISA results for 2017 ((OD sample/OD negative control) × 100) ranged from 15.35 to 64.05 with a mean of 84.64 (SD: 15.354, *n* = 40), while the results from 2018 and 2019 ((OD sample − OD negative control)/(OD positive control − OD negative control) × 100) ranged from 0.05 to 21.81 with a mean of 1.85 (SD: 2.094, *n* = 241). The results for all the ELISA tests are given in Appendix A.

Antibodies directed against pestivirus were detected in two of 280 reindeer (0.7%), both hunted in Area 2 in 2017 [43]. All animals sampled in 2018 and 2019 and tested for pestivirus antibodies were sero-negative. Samples from fifteen reindeer were classified as doubtful according to the ELISA evaluation criteria (Table 3). They were all retested and remained classified as doubtful.

Antibodies against MCFV were detected in two of 280 animals (0.7%), both from 2018. One was an adult male from Area 1 with I% = 45.5 and the other was a barren female from Area 2 with I% = 35.7% (Figure 2, Table 3, Appendix A) in the MCFV cELISA. PCRs conducted on leucocyte samples (“buffy coat”) from these two animals revealed inconsistent and faint bands from one of the animals in both PCRs. The PCR-products were cloned and sequenced but with unspecific results that did not support the serology results. The other seropositive animal and the eight seronegative reindeer were consistently negative by the two PCRs (results not shown).

Parapoxvirus-specific DNA, corresponding to the *GIF* gene, was detected by PCR in material from the nasal swab samples of two of 181 animals investigated; one 3–5-year-old lactating female from Area 5 and one adult male from Area 6, both shot in 2019 (GenBank accession numbers OM751841 and OM751840, respectively).

## 4. Discussion

This study represents the first thorough screening for potential viral pathogens in the Icelandic reindeer population. Two reindeer were seropositive against pestivirus and two reindeer had antibodies against MCFV, whereas all reindeer were seronegative against the other viruses. Furthermore, parapoxvirus-specific DNA was detected in the nose swab sample from two reindeer.

Two animals had antibodies against pestivirus, as previously reported in a geographically broader screening of reindeer from Iceland, Norway, Sweden, Finland, and Russia [43]. The two sero-positive animals were retested two times and in duplicate in two different laboratories, confirming the positive ELISA results. In a VNT against Pestivirus A, one of these two samples tested positive again, whereas the other showed non-conclusive results due to toxic effects on the cells [43]. In addition, samples from 15 reindeer were classified as doubtful after being retested. There are no known records of pestivirus being present in Iceland, in reindeer or livestock, in recent times.

Serological screenings of Norwegian reindeer sampled in 2006–2008 from throughout Finnmark County (*n* = 3339) and of reindeer from reindeer herding district 16 in particular (*n* = 588), which is close to Avjovarre from where the animals were exported to Iceland in 1787, revealed a seroprevelance of 12.5% and 0.7%, respectively [21]. These results suggest that pestivirus is enzootic in semi-domesticated reindeer in Finnmark today and that in spite of being 230 years apart the virus may have been imported to Iceland with the reindeer from Norway. Alternatively, pestivirus infections may have occurred unnoticed in cattle or sheep in Iceland and spilled over to reindeer, although this may not be very likely.

It is challenging to suggest the potential impact of pestivirus infections in Icelandic reindeer. In semi-domesticated reindeer in Fennoscandia, a seroprevalence reaching 50% may be found in some herds, but there are no associated records of high incidence of poor reproduction (abortion, stillbirths, weak borne calves) or young animals with immunosuppression and poor fitness, which would be expected findings based on the impact of pestivirus A (former BVDV) infections in cattle [49]. However, it is also important to keep in mind that these clinical manifestations may be hard to detect in remote populations of free ranging wildlife. An experimental inoculation of reindeer with pestivirus A induced diarrhea and nasal discharge, as well as viraemia and seroconversion, indicating an active infection [50]. Isolation of pestivirus from reindeer has been reported only once. The isolate (V60-Krefeld, Reindeer-1) was obtained from a naturally infected reindeer at Duisburg Zoo, Germany, having severe diarrhea [51]. Genetic studies revealed a close relationship to Pestivirus D (i.e., former border disease virus type 2; BDV-2) isolated from German sheep [52]. This finding was supported by the VNT analyses of seropositive reindeer from Finnmark, Norway, against a panel of ruminant pestiviruses (two isolates of Pestivirus A and one isolate of Pestivirus D), demonstrating a stronger neutralizing capacity against Pestivirus D (former BDV-2) as compared to Pestivirus A [21].

The two pestivirus seropositive reindeer were both 3–5-year-old; a male (82 kg CW) and a female (42 kg CW), but had no clinical signs indicating pestivirus infection. Based on the low seroprevalence found in this study, even if counting in the 15 animals classified as doubtful and the lack of clinical signs suggesting pestivirus infections, we have no reason to believe that pestivirus infections represents a threat to the reindeer population in Iceland. Furthermore, since other studies have indicated that reindeer may be hosting their own pestivirus variant [21,52], we believe that reindeer do not necessarily represent a hazard for spillover to livestock such as sheep. However, the virus that is present among reindeer in Iceland may have such characteristics and should be further characterized.

Two reindeer were classified as seropositive (ELISA) for the MCFV-group, with both animals being scored as clearly positive (i.e., I% ≥ 25) and being the only animals with a score above 22. However, these results could not be supported by identifying MCFV-specific DNA in samples from the seropositive animals. This indicates a very low seroprevalence against viruses of the MCFV group among reindeer in Iceland, or that the two seropositive animals represent false positives in the ELISA.

The MCFV seroprevalence of 0.7% is somewhat lower than the seroprevalence of 3.5% that was found in semi-domesticated reindeer (*n* = 3339) in Finnmark county [20]. This low seroprevalence, associated with the absence of clinical cases reported in the country, suggests that MCFVs have limited health impact on the reindeer population in Iceland. In addition, the presence of a novel gammaherpesvirus, tentatively called Rangiferine gammaherpesvirus 1, which may be host-specific to reindeer, has been recently reported in Norwegian semi-domesticated and wild reindeer [53]. Phylogenetic analysis of this virus, based on sequences from DNA polymerase and glycoprotein B genes, suggested that it is not a MCFV, and therefore not expected to have the 15A epitope targeted by the cELISA used in this study. However, further investigation is necessary to examine possible antibody cross-reactivity and to address which gammaherpesvirus these animals have been exposed to and the possible impact of such infections.

The humoral immune response against ORFV is characterized as short-lived in sheep [54]. Since we assumed the same would be the case for reindeer, and since we had access to reindeer heads for investigation of clinical signs of CE, we chose to investigate swab samples from oral and nasal mucosal membranes for virus shedding rather than investigate for antibodies in blood. No clinical signs resembling CE were registered when inspecting the muzzle and oral mucosa of 220 reindeer heads during this study. Nevertheless, parapoxvirus-specific DNA was amplified from nasal swab samples obtained from two reindeer (2019) from Area 5 and 6 (Figure 2). The generated sequences indicate ORFV, which assumingly reflects introduction of the virus to reindeer via sympatric grazing sheep [25] rather than being enzootic in the reindeer population. The finding of parapoxvirus in clinically healthy animals is supported by previous investigations, reporting parapoxvirus-specific DNA detected by PCR in reindeer carcasses in Finnmark county, with no indications of clinical CE [55], and in apparently healthy animals from Finnmark, Nordland, and Trøndelag counties, Norway, as indicated by Next Generation Sequencing [34]. In spite of these findings, clinical CE has been recorded only a few times in Norway [19], indicating that the presence of the virus alone may not elicit a disease outbreak. With only one major outbreak being registered among reindeer in Iceland, in 2016 [22], these data suggest that ORFV may be regarded as an opportunistic infection in Icelandic reindeer, which can cause a clinical outbreak should the right circumstances occur. The outbreak recorded in 2016, affecting at least 31 reindeer, demonstrated that the virus may spread among free-ranging reindeer, affecting adults (e.g., udder skin infections) and their calves. Furthermore, ORFV may represent a human health risk for hunters and people handling reindeer carcasses.

Interestingly, we were not able to detect antibodies against reindeer alphaherpesvirus. CvHV2, or similar alphaherpesviruses, seems to be enzootic in most reindeer and caribou populations that have been investigated [19]. A serological screening of 3062 semi-domesticated reindeer, representing 14 reindeer herding districts in East and West Finnmark County, revealed a seroprevalence of 48%, while in the reindeer herding district 16, close to Avjovarre from where the reindeer were exported to Iceland in 1787, a prevalence of 54% (2009; *n* = 590) was reported [36].

Although separated in time for about 230 years, we expected that some of the tested Icelandic reindeer would host the life-long alphaherpesvirus infection. We do not know the age of the animals that were imported but may assume they were all sexually mature. If CvHV2 was enzootic at that time, and at the same prevalence level as today, it would have been expected that about 15–16 animals would have been infected at the time of shipping. Thus, the selection, translocation, and the new habitat and living conditions for the host may somehow have represented a bottleneck affecting the genetic variation of the virus [56] so it did not manage to prevail in the population. Alternatively, alphaherpesvirus might not have been present in the reindeer herd in Finnmark at the time of export, being introduced and becoming enzootic there later. Having inspected and sampled the mucosal membranes of 220 reindeer heads during this study period, we found no animals with IKC or other clinical signs such as lesions in the oral mucosa or the skin indicating alphaherpesvirus infections. Since alphaherpesvirus is a common pathogen in reindeer and a facilitator for secondary bacterial infections, the lack of alphaherpesvirus infections among Icelandic reindeer is assumed to be very beneficial for their health status.

No antibodies were detected against BTV or SBV; BTV is thought to be transmitted by biting midges [57], whereas SBV has been mostly isolated from biting midges (*Culicoides* spp.; Diptera, Ceratopogonidae), but also to a lesser extent from mosquitos [58]. Mosquitos are absent in Iceland, whereas four species of midges have been observed, with three belonging to genus *Simulium*, with *Simulium vittatum* (black fly, Icelandic: *mývarg*), a North American species, as the most common. Black flies (*Simulium bivittatum*) are potential vectors for vesicular stomatitis virus in North America [59], but whether this or the other species of midges in Iceland can be a vector for BTV or SBV is unknown.

Traditionally, the study of viral diseases has focused on human health and diseases of economic impact for the livestock industry. However, it has become more and more evident that viruses circulating in wildlife may not only affect those populations, but also have impacts in a One Health perspective, representing links to the human and livestock interface. The findings in this study indicates that Icelandic reindeer do not currently represent a hazard for transmission of infectious agents and diseases to livestock but suggest that sympatric sheep may be a source of parapoxvirus infections that can cause CE outbreaks in reindeer and zoonotic infections. Furthermore, the exposure to pestivirus and MCFV should be further investigated. Increased animal densities and contact between sheep and reindeer may, in the future, may impact the favorable health situation for reindeer in Iceland.

## 5. Conclusions

The presence of antibodies against pestivirus and MCFV, as well as the finding of parapoxvirus-specific DNA in nose swab samples indicates exposure but a low prevalence of these reindeer pathogens. These findings are supported by the fact that no epizootics or disease outbreaks have been registered in this population in recent times, with the exception of an outbreak of CE in 2016, caused by ORFV assumingly being transmitted from sheep. We found no indication of exposure of reindeer to CvHV2, which is an enzootic reindeer pathogen in Fennoscandia and in other *Rangifer* populations. In conclusion, enzootic viral infections currently seem to have restricted impact on the health of Icelandic reindeer.

## Figures and Tables

**Figure 1 viruses-15-00317-f001:**
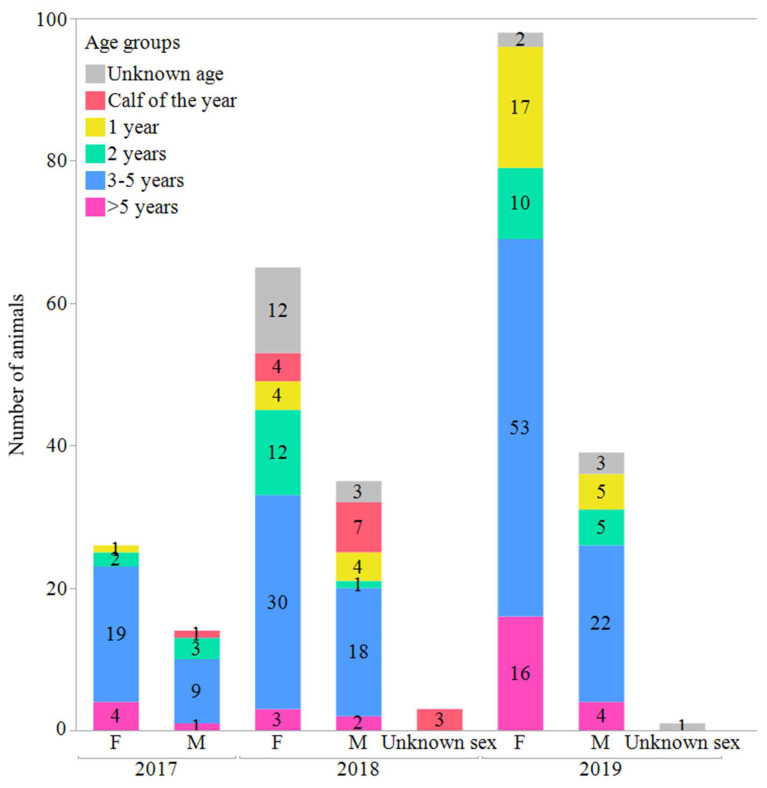
Distribution of wild Icelandic reindeer (*n* = 281) sampled in 2017–2019 and tested for exposure to selected viral infections.

**Figure 2 viruses-15-00317-f002:**
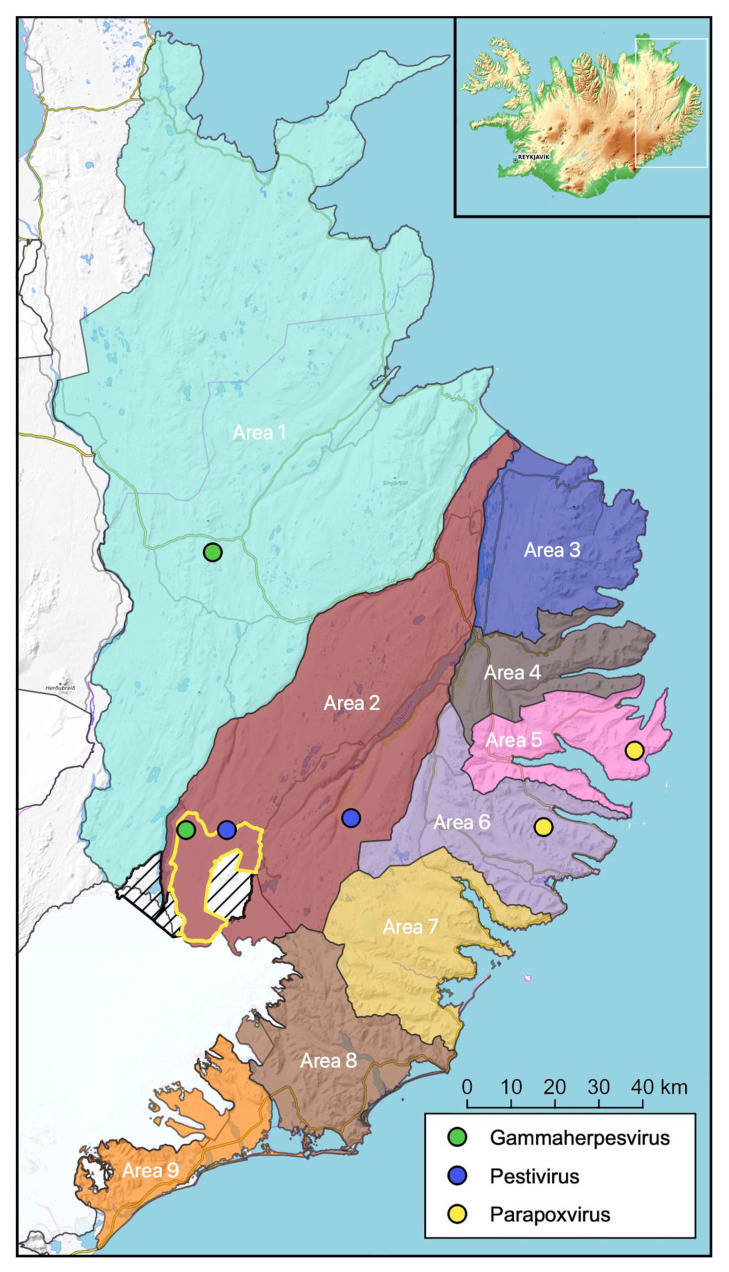
Icelandic wild Eurasian tundra reindeer (*n* = 281), all being descendants from 35 animals imported from Norway in 1787, were sampled and tested for exposure to potential reindeer viral pathogens. Two reindeer with anti-pestivirus antibodies are indicated with blue dots (Area 2), two reindeer with antibodies against gammaherpesvirus (MCFV-group) are indicated with green dots (Area 1 and 2) and two reindeer shedding parapoxvirus on their nasal mucosal membrane are indicated with yellow dots (Area 5 and 6). All animals were seronegative for alphaherpesvirus, bluetongue virus, and Schmallenberg virus. The yellow line in Area 2 indicates the national park and the two shaded areas indicate protected areas around the Snæfell mountain.

**Table 1 viruses-15-00317-t001:** A total of 281 Icelandic wild Eurasian tundra reindeer (*Rangifer tarandus tarandus*) (2017–2019) with all the nine hunting areas represented were investigated for antibodies against a panel of potential viral pathogens.

Hunting Area	2017 ^1^	2018 ^1^	2019	Total
1	10	26 (1)	22	58 (1)
2	25 (1)	39 (7)	42	106 (8)
3	1	6 (1)	12	19 (1)
4	1	2	3	6
5	1	3	21	25
6	1	2	7	10
7	1	14 (3)	16	31 (3)
8	0	0	8	8
9	0	9 (1)	7	16 (1)
Unknown	0	2	0	2
Total	40 (1)	103 (14)	138	281 (15)

^1^ Numbers for 2017 and 2018 include 1 and 14 calves of the year, respectively. The calves are included in the total number, and calf numbers per district are specified in parenthesis. All other animals were adults (14 months or older).

**Table 2 viruses-15-00317-t002:** Serological tests used to investigate wild Eurasian tundra reindeer (*Rangifer t. tarandus*) from Iceland (2017–2019) for antibodies against viral pathogens.

Viral Pathogen	Test and Producer	Test Principle	Antigen	References
Alphaherpes	SERELISA BHV-1 gB Ab Mono Blocking. Synbiotics, France	Blocking ELISA	BoHV1 glycoprotein B	[36]
Gammaherpes	Non-commercial ^1^	Competitive-inhibition ELISA	MCFV (AlHV-1) glycoprotein	[37,38]
Pesti	SERELISA^®^ BVD p80 Ab Mono Blocking. Synbiotics, France	Blocking ELISA	p80/p125	[39]
Bluetongue	ID Screen^®^ Bluetongue Competition. ID Vet, Grabels, France	Competitive ELISA	Recombinant, VP7	[40]
Schmallenberg	ID Screen^®^ Schmallenberg virus competition multispecies ^2^. ID Screen^®^ Schmallenberg Indirect multispecies ID Vet, Grabels, France	Competition ELISA Indirect ELISA	Recombinant SBV antigen	[41]

^1^ Animal Disease Research Unit, US Department of Agriculture—Agricultural Research Service, Washington State University, Pullman, Washington, USA. ^2^ The competition ELISA was used for the 2017 samples, whereas the Indirect ELISA was used for the 2018–2019 samples. See text for details regarding evaluation criteria.

**Table 3 viruses-15-00317-t003:** Summary of results for each serological screening and year, indicating seropositive/tested (percentage).

	2017	2018	2019	Total
Alphaherpesvirus	0/40 (0%)	0/103 (0%)	0/138 (0%)	0/281 (0%)
Gammaherpes MCFV-group	0/40 (0%)	2/103 (1.9%)	0/137 (0%)	2/280 (0.7%)
Pestivirus	2(2) ^1^/40 (5–10%)	0(12) ^1^/102 (0–11.8%)	0(1) ^1^/138 (0–0.7%)	2(15) ^1^/280 (0.7–6.1%)
Bluetongue virus	0/40 (0%)	0/103 (0%)	0/137 (0%)	0/281 (0%)
Schmallenberg virus	0/40 (0%)	0/103 (0%)	0/137 (0%)	0/281 (0%)

^1^ For pestivirus, serum samples/animals that were classified as doubtful by the test are given in parenthesis, resulting in seroprevalence as a range; the lower seroprevalence if doubtful samples were regarded as negative and the higher seroprevalence if doubtful samples were regarded as positive. For the other ELISA kits operating with a doubtful score (i.e., Alphaherpesvirus and Schmallenbergvirus, SBV), none of the samples were classified as doubtful.

**Table 4 viruses-15-00317-t004:** A summary of sero-positive and PCR-positive reindeer indicating the investigated pathogen, year, hunting area and place, sex, age, and carcass weight.

Pathogen	Year	Hunting Area ^1^	Place	Sex	Age	Carcass Weight (kg) ^2^
Pestivirus (antibodies)	2017	2	Fljótdalsheiði	F	3–5	42
	2017	2	Flatarheiði	M	3–5	82
Gammaherpesvirus (MCFV; antibodies)	2018	1	Kollseyra	M	3–5	105
	2018	2	Vestaradrag	F	1	33
Parapoxvirus (virus-specific DNA; PCR)	2019	5	Karlsstaðatindur/Ímadalur	F	3–5	47
	2019	6	Stöðvarfjörður	M	Adult	108

^1^ Hunting area, as depicted in Figure 2. ^2^ Carcass weight: weight of the animal in kg after bleeding and when the head, internal organs, skin, and lower parts of the legs have been removed.

## Data Availability

The dataset used and analyzed during the current study is provided as Appendix A. Raw data (ELISA read-outs) are available from the corresponding author on reasonable request.

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
