# Peer review of "A Screening for Virus Infections among Wild Eurasian Tundra Reindeer (Rangifer tarandus tarandus) in Iceland, 2017–2019"

_viruses, 2023, doi:10.3390/v15020317_

Round 1

Reviewer 1 Report (Previous Reviewer 3)

In the revised version of the manuscript "A screening for virus infections among wild Eurasian tundra reindeer (Rangifer tarandus tarandus) in Iceland, 2017-2019 " by Tryland M. et al., Manuscript ID: viruses-2128215-peer-review-v1, the authors have satisfactorily addressed all my comments and suggestions. I have no other specific comments.

Author Response

Reviewer is satisfied with Revision # 1. No further comments.

Reviewer 2 Report (Previous Reviewer 2)

In this manuscript, Tryland et al. survey a hunted population of 281 wild Eurasian tundra reindeer from 9 areas across Iceland for the presence of antibodies against various viral pathogens or parapoxvirus DNA as evidence of viral infection to elucidate the prevalence of viral pathogens in an introduced reindeer population. The authors cover a wide range of viral genera in their survey and identify low prevalence or absence of the sought for viral pathogens. The manuscript is clearly written with very minimal text errors.

In this resubmission the authors have made multiple adjustments to improve the manuscript, including some restructuring. Figure 1 has been revamped to be much easier to interpret. The methods have been rewritten to include much more detail and look to be much more reproducible. However, Figures 1 and 2 could be moved to the Results section to bulk this up a bit (survey information is still useful to report here), especially as results are communicated in the legend of Figure 2, inappropriate for a methods section if it is kept there. Figure 1 aesthetics should be improved to match the higher quality of Figure 2.

One of my main questions in a previous review was whether or not a metagenomic analysis could be performed on the samples provided, or at least have the strength/limitations of that approach discussed appropriately. This is lacking from the manuscript at present, and no rebuttal/response to reviewer comments document has been made available for me to understand the author’s response.

In summary, though the article is well written with acceptable technical approaches, I recommend this manuscript would be more appropriate as a Communications-type article rather than a research article because of its brevity and limited impact. The length of the introduction and discussion do not match the limited results presented, and instead read more as a review of the relevant literature without introducing a significant finding and step forward with respect to management of the 25% unknown mortality. These sections should be condensed and the manuscript reformatted as the shorter ‘Communication’ submission if published within Viruses.

Round 2

Reviewer 2 Report (Previous Reviewer 2)

I apologise to the authors - the initial rebuttal letter had not been made available to me when reviewing the revised manuscript (it now has been) and I can see multiple comments were addressed and explained within.

I am satisfied with the current revisions made by the authors and recognise the wish to maintain the format as a full-length article.

This manuscript is a resubmission of an earlier submission. The following is a list of the peer review reports and author responses from that submission.

Round 1

Reviewer 1 Report

Major comments:

  • The authors did not respect the Viruses submission format. Is this an article or short communication? There is no line in the manuscript presented which makes it a pain to review and reference specific items.

  • Introduction:
    • Page 5: Study rationale is rather weak. All together, the unexplained reindeer deaths that the authors are trying to explain is 25% of 252 = 63 deaths spread across 13 years… That’s less than 5 deaths a year…
    • Page 6: How pathogenic are these viruses to reindeer?
    • Additional introduction about the situation in original population of reindeer from Norway would help compare and maybe provide some more relevance. Is there a large number of reindeer death accounted to viral infection in wild and semi-domesticated population? Is this an actual source of concern for the population or the industry?
    • Are these pathogens of concern regarding meat consumption?

  • Method:
    • I’m a little worried about sample quality due to the nature of study because of the timeline (across many years) and through sampling carcasses. Do the authors test any positive control antibody with their ELISA assay contained in the reindeer blood to validate the samples were usable (common pathogens for instance)?

  • Discussion:
    • The discussion does address some of my above questions, but the authors could use some of that information to improve it.
    • Discussion right now is very descriptive and could be improved.
      • How can Icelandic populations of reindeer can be preserved? Is agricultural practice a danger for introduction of related pathogens as seen which ORFV and sheep (page 17) for instance?

Minor comments:

  • Introduction:
    • Introduction of reindeer populations need references.

  • Discussion:
    • There are some typos that need to be addressed, another round of proof reading might be necessary.

Reviewer 2 Report

In this manuscript, Tryland et al. survey a hunted population of 280 wild Eurasian tundra reindeer from 9 areas across Iceland for the presence of antibodies against various viral pathogens or parapoxvirus DNA as evidence of viral infection to elucidate the prevalence of viral pathogens in an introduced reindeer population. The authors cover a wide range of viral genera in their survey and identify low prevalence or absence of the sought for viral pathogens. The manuscript is clearly written with very minimal text errors.

Unfortunately, there are structural issues with the format of the manuscript that should be addressed to improve the article. The map in Figure 1 should be moved to the methods, with all experimental outcomes (seroprevalence etc.) stripped from the legend. This should not be present in the introduction. Further, Table 1 adds little to the introduction and instead could be combined with Table 2 within the Methods section to clearly indicate the population studied. Much of the introduction should be summarised and condensed, with a lot of the text regarding the viruses tested to be placed later in the manuscript within the discussion so as to better contextualise the results. That the introduction is so much longer than the results as presented lends to an interpretation that the results are less significant to the field than deserved.

The main issue with this manuscript is the presentation of the results. Importantly, a single table of summary data is not enough for publication within the journal. At minimum supplementary data for each assay should be available (or have been made available at the review stage), with key data presented within the body of the Results section. Without analysed data for each assay, a reader cannot confirm for themselves whether the conclusions agree with the data. For publication, I believe that an overhaul of the results section would be required, with table/figure data demonstrating the outcome for each kit assay/PCR. Confidence intervals/means etc. are all absent and so it's hard to gauge the quality of the datasets. 

Further, at multiple points in the discussion, the authors identify that 'based on these data alone' true prevalence is hard to determine, with mention of follow-up studies needed to truly understand the impact of these viruses. For this reason, I feel this manuscript would be more appropriate as a Communications-type article rather than a research article because of its brevity and limited impact, unless those further studies are folded in in a revised version of the manuscript.

Particular queries:

  1. Hunting area 1 is by far the largest - does it also have a larger population and hence more hunted animals? Table 2 presents Area 1 as having fewer animals studied than the smaller Area 2. Is there a fair reason why area and proportionality in your study aren't balanced?
  2. How does the carcass weight influence the results in Table 4?
  3. The body text referring to the results of pestivirus testing indicate both positive animals were hunted in Area 2 in 2017 (but Table 4 suggests it was Area 5 and 6). Please correct whichever is erroneous. Has it been mixed up with the Areas listed for parapoxvirus?
  4. Though most viruses investigated in this study were surveyed by serological testing, parapoxvirus was assessed by PCR, an assay that will only identify current infections. Can you address why serological testing was not performed to identify separately the prevalence of this virus in the population rather than just identifying active infection? This assay stands out for addressing a very different aim.
  5. The presentation of Figure 2 is too complex to pull simple information from. As an option, it may be better to present these as proportion of a whole plot (either columns adding to 100% or pie charts). Present male vs female and the five ages in various colours with three total columns (2017/2018/2019). Further, structurally this could be moved to the Results section.
  6. In the methods, could a brief description of the requirements for each assay be provided: how much serum etc. for each serological assay, what are your primer sequences for PCR etc.? I acknowledge you've referenced previous use of the B2L and GIF PCR, so perhaps the primers aren't needed, but this data output is DNA gel electrophoresis/sequencing and none of this kind of data is available to reviewers/your readers.
  7. Is a metagenomic approach possible in remaining samples of swabs/blood? Could you survey databases of viral sequences against an unbiased sequencing screen to determine presence of more viruses than tested in this panel?
  8. What antigen is detected for the Gammaherpes non-commercial serological test? It's absent from Table 3.

Reviewer 3 Report

The manuscript "A screening for virus infections among wild Eurasian tundra reindeer (Rangifer tarandus tarandus) in Iceland, 2017-2019 " by Tryland M.  et al., Manuscript ID: viruses-1692070, describes a viral epidemiological evaluation of Eurasian tundra reindeer. In this study, the authors evaluated the presence of antibodies for alphaherpesvirus, gammaherpesvvirus (malignant catarrhal fever viruses, MCFV), pestivirus, blue tongue virus, and Schmallenberg virus in 280 blood samples. Furthermore, they investigate 181 nasal/oral swab samples for the presence of parapoxvirus-specific DNA. All samples were collected during the hunting seasons in 2017, 2018, and 2019. Antibodies against pestivirus were detected in samples from two animals hunted in 2017, and antibodies against MCFV were seen in the other two reindeer sampled during the hunting season in 2018.  No antibodies were detected against the other viruses tested. Parapoxvirus-specific DNA was detected in nasal swab samples from two animals collected in 2019. After careful evaluation, I have some significant concerns regarding the publication of this article in the journal Viruses.

Herein you can find the comments regarding the manuscript:

In the section “Materials and methods,” the authors use just serologic tests to screen the blood samples to evaluate the exposure of Eurasian tundra reindeer to different viruses, including alphaherpesvirus, gammaherpesvvirus (malignant catarrhal fever viruses, MCFV), pestivirus, blue tongue virus, and Schmallenberg. Why didn't the authors use molecular protocols other than serological tests? The molecular analysis could have provided more information on the viral strains found in this study. Moreover, the choice of viruses tested in this study is ambiguous.

Can Eurasian tundra reindeer be considered potential reservoirs of pathogens causing zoonotic infections?

The “Discussion” and the “Conclusion” should be better structured to stress the importance of studying the virome of wild animals. Virus surveillance in wild and domestic animals, using well-described conventional and molecular methods and NGS technologies, could be crucial for monitoring and promptly characterizing emerging and re-emerging zoonotic viruses, providing a baseline of viral diversity and circulation, helpful in dealing with future infectious emergencies.

Although this manuscript provides preliminary data that should be further investigated, the manuscript could be worth publishing as Brief Report.